
**Urban anomalies in response to rainstorms based on smartphone location data: a**
**case study of eight cities in China**
**Jiawei Yi[1,2], Yunyan Du[1,2*], Fuyuan Liang[3], Tao Pei[1,2], Ting Ma[1,2], Chenghu Zhou[1,2]**
[1]State Key Laboratory of Resources and Environmental Information System, Institute of
Geographic Science and Natural Resources Research, Chinese Academy of Sciences, Beijing, China
[2]University of Chinese Academy of Sciences, Beijing, China
[3]Department of Geography, Western Illinois University, Macomb, IL, USA
*Corresponding author: duyy@lreis.ac.cn
**Abstract**
This study explored city residents' collective geo-tagged behaviors in response
to rainstorms using the number of location request (NLR) data generated by
smartphone users. We examined the rainstorms, flooding, NLR anomalies, as well as
the associations among them in eight selected cities across the mainland China. The
time series NLR clearly reflects cities' general diurnal rhythm and the total NLR is
moderately correlated with the total city population. Anomalies of NLR were
identified at both the city and grid scale using the S-H-ESD method. Analysis results
manifested that the NLR anomalies at the city and grid levels are well associated with
rainstorms, indicating city residents request more location-based services (e.g. map
navigation, car hailing, food delivery, etc.) when there is a rainstorm. However,
sensitivity of the city residents' collective geo-tagged behaviors in response to
rainstorms varies in different cities as shown by different peak rainfall intensity
thresholds. Significant high peak rainfall intensity tends to trigger city flooding, which
lead to increased location-based requests as shown by positive anomalies on the
time series NLR.

Keywords: urban anomaly; rainstorm disaster; human response; rainfall intensity
threshold; anomaly score

**1 Introduction**
Global climate change is making rainfall events heavier and more frequent in
many areas. Powerful rainstorms may flood a city once the rainfall exceeds the
discharge capacity of a city's drainage system. Inundation of cities' critical
infrastructures and populated communities tends to disrupt urban residents' social
and economic activities and even cause dramatic life and property losses
(Papagiannaki et al. 2013; Spitalar et al. 2014; Liao et al. 2019). Floods nowadays are
the most common type of natural disaster, which poses a serious threat to the safety
of life and property in most countries (Alexander et al. 2006; Min et al. 2011; Hu et al.
2018). According to the released survey in the Bulletin of Flood and Drought


Disasters in China, more than 104 cities were struck by floods in 2017, affecting up to
2.18 million population and causing over 2.46 billion US dollars direct economic
losses (China National Climate Center 2017).

4        The impacts of a rainstorm are usually evaluated with respect to the interactions

among rainfall intensity, the population exposure, the urban vulnerability, and the
society coping capacity (Spitalar et al. 2014; Papagiannaki et al. 2017). The rainfall
intensity that may trigger flood disasters has been extensively investigated and
many studies have examined the relationship between rainfall intensities and social
responses (Ruin et al. 2014; Papagiannaki et al. 2015; Papagiannaki et al. 2017).
Nowadays the peak rainfall intensity is widely used to determine the critical rainfall
threshold for issuing flash flood warnings (Cannon et al. 2007; Diakakis 2012; Miao
et al. 2016).

13       The population exposure refers to the spatial domain of population and

properties that would be affected by a rainfall hazard (Ruin et al. 2008). Gradual
increase in the proportion of population living in urban areas due to urbanization
makes more people exposed and vulnerable to urban flash floods, posing great
challenge to flood risk reduction (Liao et al. 2019). Reduction of vulnerability
therefore becomes critical in urban disaster mitigation. Vulnerability is usually
assessed by comprehensively considering related physical, social, and
environmental factors (Kubal et al. 2009; Adelekan 2011; Zhou et al. 2019), and
their dynamic characteristics across space and time (Terti et al. 2015).

22       Coping capacity reflects the ability of a society to handle adverse disaster

conditions and it is one of the most important things to consider in disaster
mitigation (UNISDR 2015). The coping capacity is usually evaluated by examining the
human behaviors in response to disasters, which are mainly collected by
post-disaster field investigation and questionnaires (Taylor et al. 2015). Such
conventional approaches only provide limited samples that may not be able to fully
and timely reflect disaster-induced human behaviors. Recently, researchers have
learned the advantages of using unconventional data sets such as insurance claims
(Barberia et al. 2014), newspapers (Llasat et al. 2009), and emergency operations
and calls (Papagiannaki et al. 2015; Papagiannaki et al. 2017) to quantify the coping
capacity.

33       The growing use of smartphones and location-based services (LBS) in recent

years has generated massive geospatial data, which could be used to infer the
collective geo-tagged human activities. The geospatial data thus provides a new
perspective to study normal urban rhythm in regular days (Ratti et al. 2006; Ma et al.
2019) and abnormal human behaviors in response to emergencies (Goodchild &
Glennon 2010; Wang & Taylor 2014; Kryvasheyeu et al. 2016). Bagrow et al. (2011)



found the number of phone calls spiked during earthquake, blackout, and storm
emergencies. Dobra et al. (2015) explored the spatiotemporal variations in the
anomaly patterns caused by different emergencies. Gundogdu et al. (2016) reported
that it is possible to identify the anomalies inflicted by emergencies or
non-emergency events from mobile phone data using a stochastic method. In
addition to the afore-mentioned applications, more studies are needed to explore
the full potential of the mobile phone data in terms of revealing human collective
behaviors, particularly in response to hazards and emergencies.

9       This study explored the urban anomalies and their variations in response to

rainstorms using the NLR requests from smartphone users. We selected eight
representative cities in the mainland China to examine how urban residents response
to typical summer rainstorms in different regions. The anomalies of LBS requests
caused by rainstorms were identified using a time series decomposition method and
then described by multiple indices, which are used to study how rainstorms affect
geo-tagged human behaviors collectively. The rest of the paper is organized as
follows. Section 2 introduces the selected cities and the smartphone NLR dataset.
Section 3 presents the anomaly detection and description methods. Section 4
provides the analysis results including rainfall statistics, normal rhythms, and
rainstorm-triggered anomalies in the selected cities. Section 5 concludes the study
and discusses the future work.
**2 Materials**
**2.1 Study area**

24       We selected eight representative cities across the mainland China for this study

(Fig. 1). Two cities were selected from each region except the northwestern and
southwestern China (Table 1). The eight cities vary significantly with respect to their
total population, footprint areas, and urbanization rate. In this study, the footprint of
a city is composed of the grids that have an hourly number of location requests (NLR)
no less than the median of the daily NLR time series of that grid over the whole
month, i.e., the grids with at least one NLR every hour in average.

31       Haikou and Zhuhai are located in southern China which has mean annual

precipitation between 1600 mm and 3000 mm. Among the eight cities, Zhuhai is the
least populated city but with the highest urbanization rate. In central China, we
selected Hefei and Xiangyang, which have mean annual precipitation between 800
mm and 1600 mm. Two cities, Lanzhou and Hengshui, were selected from a
semi-humid region in northern China with mean annual precipitation between 400
mm and 800 mm. Hengshui has the largest footprint area but the least urbanization


rate among the cities. Harbin and Jilin are located in the Northeastern China. The
mean annual precipitation of Harbin and Jilin ranges from 400 mm to 800 mm and
between 800 mm and 1600 mm, respectively. Harbin is the most populated among
the eight cities.

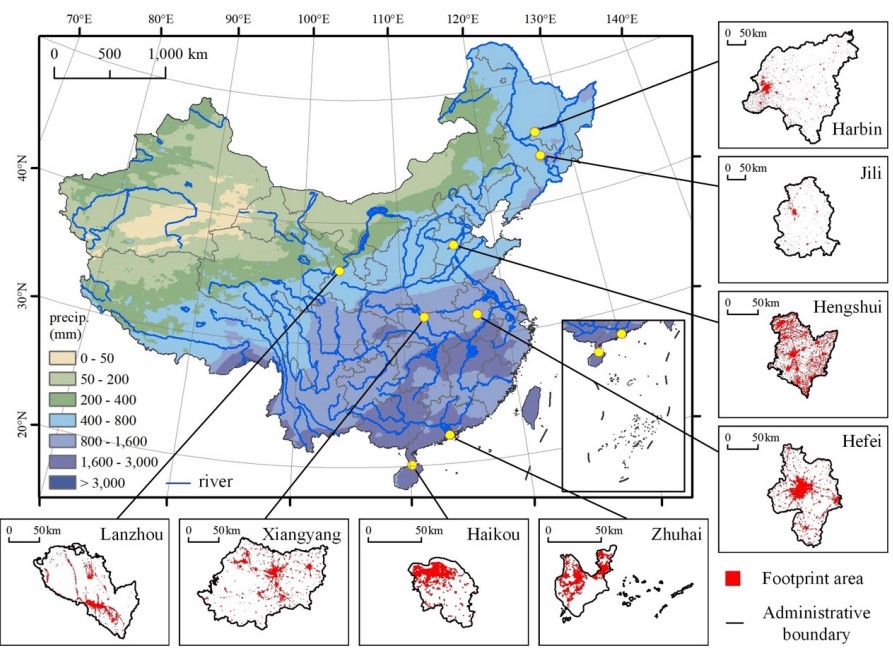

Figure 1 A map showing the geographic locations, annual precipitation, and
footprints of the eight cities in this study.
Table 1 Statistics of the cities

| Region | City | Population (10^4) | Footprint area (km²) | Urbanization rate (%) |
|---|---|---|---|---|
| Southern China | Haikou | 227.21 | 625 | 78.21 |
| | Zhuhai | 176.54 | 567 | 89.37 |
| Central China | Hefei | 796.50 | 1927 | 73.75 |
| | Xiangyang | 565.40 | 1817 | 59.65 |
| Northern China | Lanzhou | 372.96 | 1219 | 81.02 |
| | Hengshui | 446.04 | 2997 | 50.60 |
| Northeastern China | Harbin | 1092.90 | 2083 | 64.50 |
| | Jilin | 415.35 | 704 | 52.80 |


**2.2 Data collection**

The smartphone location data was obtained from the Tencent big data portal (https://heat.qq.com/). The portal provides location request records of the global smartphone users via the Tencent Map API. A location request record is generated when a smartphone user requests any LBS, which include but are not limited to navigation, car hailing, food and merchandise delivery, or social media check-ins. The portal releases the number of location requests per 0.01×0.01 regular grid for every 4-5 minutes. Ma (2019) compared the NLR dataset with visitor numbers in a few places and confirmed that the NLR data is a good proxy of dynamic population changes. We collected the NLR data of the grids within the administrative boundaries of the eight cities from August 1 to 31, 2017.

This study used the Version 05B GPM/IMERG 30-minute precipitation dataset (Huffman et al. 2018), which has a spatial resolution of 0.1×0.1 degrees. This dataset has been evaluated and widely used (Wang et al. 2017; Zhao et al. 2018; Su et al. 2018). The news reports about the flooding events in the eight cities were mainly collected from the Chinese mainstream online media, including Xinhuanet, Ecns.cn, Sohu, etc.

**3 Methods**

**3.1 Time series anomaly detection**

The smartphone location request record can be represented by a series of spatial points: $\{(x_i, y_i, Ts_i)\}$, $i=1,2,…,n$. Each point contains its geographic coordinates ($x$, $y$) and a time $(T)$ when the LBS is requested. The NLR was then aggregated to time series per grid or per city as illustrated below.

At the city level, a time series hourly NLR was established by adding up all location requests of the grids within the footprint area of that city. The magnitudes of the NLR in different cities vary significantly due to the different numbers of smart phone users. To make the NLR in different cities comparable, we normalized the NLR using the median-interquartile normalization method, which is more robust to anomalies than other common approaches using sample mean and standard deviation (Geller et al. 2003).

We employed the S-H-ESD method (Vallis et al. 2014) to detect anomalies from the time series NLR, which can be represented by the following additive model

$$Ts=T+S+R \qquad\qquad (1)$$

where $T$, $S$, and $R$ denote the trend, seasonality and residual components in the time series data, respectively. The S-H-ESD method assumes that the trend and the seasonality would not be significantly disrupted by rapid-evolving events that last for



only a few hours. Two major steps are involved in the method. First, it uses the
piecewise median method to fit and remove the long-term trend and then the STL to
remove seasonality (Cleveland et al. 1990). Using the STL to remove the long-term
trend would introduce artificial anomalies (Vallis et al. 2014). In this study, the
underlying trend in the time series NLR is approached using a piecewise combination
of the biweekly medians, which show little changes over the whole time series.
In the second step, the S-H-ESD method employs the generalized Extreme
Studentized Deviate (GESD) statistic (Rosner 1975) to identify the significant
anomalies in the residuals. The GESD calculates the statistic ($G$) based on the mean ($\bar{r}$)
and the standard deviation ($s$) of the observations:

$$G = \frac{max|r_j - \bar{r}|}{s} \qquad\qquad (2)$$

Given the upper bound of $u$ suspected anomalies, the GESD performs $u$
separate tests. In each test, the GESD re-computes the statistic $G$ after removing the
observation $r_j$ that maximizes $\left| r_j - \bar{r} \right|$ and then compares $G$ with the critical value $\lambda$
as defined below:

$$\lambda = \frac{(k-1)t_{1-a/(2k),k-2}}{\sqrt{k\left(k-2+t_{1-a/(2k),k-2}^2\right)}} \qquad\qquad (3)$$

where $k$ denotes the number of the observations in the time series after eliminating
a suspected anomaly in the last run, and $t_{p,d}$ represents the $p^{th}$ percentile of a $t$
distribution with a degree of freedom $d$. In this study, we set the significance level $a$
as 0.05 and the number of anomalies no more than 25% of the total observations.
Each test identifies one anomaly in the residuals when $G > \lambda$. The identified anomaly
is either a positive or negative, depending upon whether the residual is greater or
smaller than 0, respectively.

**3.2 Anomaly measures and scores**
In this study, an individual anomaly is represented with a vector,
$v=(x, y, t, obs, res)$           (4)
where $x$ and $y$ denote the coordinates of the grid centroid, $t$ denotes the observation
time, $obs$ and $res$ denote the observation and the residual ($R$ in equation 1) in the
time series. This study uses an anomaly's absolute residual to describe its unusual
deviation from its expectation.
A rainstorm disaster, once significantly impacts the cities, usually can trigger an



outbreak of NLR anomalies in multiple places across the city. To collectively
characterize the abnormal human responses, this study defines three indices: the
total number ($N_t$), the total residual ($R_t$), and the mean density ($D_t$) of the positive or
negative anomalies. The mean density is defined as follows:

$$D_t = \frac{\sum_{i=1}^{N_t} B_i}{N_t} \qquad (5)$$

where $B_i$ denotes the number of neighborhood anomalies within a Manhattan
distance of a 5-grid (~5 km) radius of the $i$th anomaly. The radius is large enough to
cover most urban facilities nearby the anomaly.

9       An anomaly score is then defined based on the afore-mentioned indices to

evaluate the city residents' responses to a rainstorm event. First, we surveyed the
hourly changes of the indices and calculated the quartiles ($Q_1, Q_2, Q_3$) and
interquartile range ($IQR$) of each index for every hour every day. The score of an
index is defined by:

$$S_{V,t} = \begin{cases} \frac{V_t - Q_1}{IQR} & , if\ V_t \leq Q_1 \\ 0 & , if\ Q_1 < V_t < Q_3 \\ \frac{V_t - Q_3}{IQR} & , if\ V_t \geq Q_3 \end{cases} \qquad (6)$$

where $V_t$ represents one of the three indices at time $t$. According to Tukey's fences
(Tukey 1977), the score is considered an outlier if its absolute value is greater than
1.5 or an extreme if it is greater than 3. The final anomaly score is the mean of the
three index scores.
**3.3 Characterization of a rainfall event**

21       In this study, we examined the city residents' responses to the rainfall events in

August 2017. The national average precipitation of this month is 124.6 mm, which is
the highest in 2017 and 21.3% more than the August average precipitation in
previous years.

25       We defined a rainfall event as a precipitation process that lasts for at least two

hours and with no rain preceding it for at least one hour. The severity of a rainfall
event is described by its duration, accumulated precipitation, and peak rainfall
intensity. The duration refers to how long a rainfall event lasts, and the accumulated
precipitation is the total precipitation received during a rainfall event. The peak
rainfall intensity ($I_d$) is widely used to estimate the possible rainfall intensity
threshold that triggers city (Cannon et al. 2007; Diakakis 2012) and is defined as
below:
$$I_d = \frac{\max\{\sum_{i=j}^{j+d-1} P_i\}}{d}, \quad j = 1, 2, \ldots, N - d + 1 \qquad (7)$$


where $P_i$ denotes the precipitation during the $i^{th}$ time interval, $N$ denotes the total
number of the time intervals in a rainfall time series, and $d$ denotes the width of the
moving time window that was used to search for the maximum accumulated
precipitation in a rainfall event. Based on the peak rainfall intensity, the August
rainfall events in the eight cities can be categorized into moderate rainstorm (0.5
mm/h < $I_1$ ≤ 4 mm/h), heavy rainstorm (4 mm/h < $I_1$ ≤ 8 mm/h), and violent rainstorm
($I_1$ ≥ 8 mm/h).
For calculation purpose, we downscaled the precipitation data to the same
spatial resolution as that of the NLR using the nearest-neighbor interpolation method.
At the city level, the rainfall of a city is defined as the total of the half-hour TRMM
precipitation within the human footprint. At the grid level, the rainfall of each grid
refers to the total precipitation received by that grid within a certain time period.
**4 Results**
**4.1 Rainfall characteristics and peak rainfall intensity thresholds**
The eight cities could be categorized into two groups in terms of the total
precipitation amount in August 2017 (Fig. 2a). The first group includes Haikou, Zhuhai,
and Hefei, with total precipitation more than 400 mm. The summer monsoon brings
plenty of water to the two coastal cities (i.e. Haikou and Zhuhai). The Typhoon Hato,
when it made landfall on August 23, further dumped 68- and 108-mm water to
Hiakou and Zhuhai, respectively. By contrast, the inland city Hefei, received 47.6%
more precipitation in 2017 than the average mainly due to a few unusual rainstorms
in August 2017 (Hydrology and Water Resource Bureau of Hefei 2018)
The second group includes all the other cities, which have less than 400 mm
precipitation in August 2017. The city Lanzhou is located in the dry northwestern
China and has the least precipitation of 250 mm. The two inland cities, Xiangyang
and Hengshui both have slightly higher precipitation of 300 mm. The precipitation of
the two northeastern cities, Harbin and Jilin, ranges between 320 and 350 mm and is
mainly brought in by the northwestern vortexes.
There are at least 15 rainstorms and two flooding events in each city. The city
Haikou, Lanzhou, and Harbin witnessed more than 20 rainstorms and about 1/4 out
of them caused serious flooding problems. The number of rainstorms in the other
cities ranges from 15 to 20 and about two to four out of them caused flooding
problems in the cities.
We identified the peak rainfall intensity threshold value that likely triggers city
flooding using the method developed by Cannon et al. (2008) and Diakakis (2012).
The method plots peak rainfall intensity of different time windows against the
corresponding rainfall duration. The flood-triggering threshold is defined as the
upper limit of the peak rainfall intensity that tends to lead to urban flooding but
actually not. As shown in Fig. 2b, for the rainfall thresholds calculated based on 0.5-,
1-, 2-, and 3-hour time window, the city ranking shows no change with an order of
Haikou, Jilin, Hengshui, Zhuhai, Hefei, Lanzhou, Harbin, and Xiangyang. The ranking
shows some fluctuations when the flooding-triggering rainfall threshold values were
calculated with a more than 3-hour time window. However, Haikou and Harbin are
always the top two cities whereas Xiangyang is the last one on the ranking list. It is
worthy to note that a rainstorm with a peak rainfall intensity over the threshold 5
mm/h would definitely trigger floods in Xiangyang. However, in Haikou, such a
threshold value is 30 mm/h. In other words, city flooding would occur in Haikou
when it is hit by a rainstorm with peak rainfall intensity over 30mm/h. In general, the
difference between the threshold values among these cities reduces with a longer
time window, indicating that the rainfall in a shorter time window is more critical to
evaluate whether a city is prone to flooding.

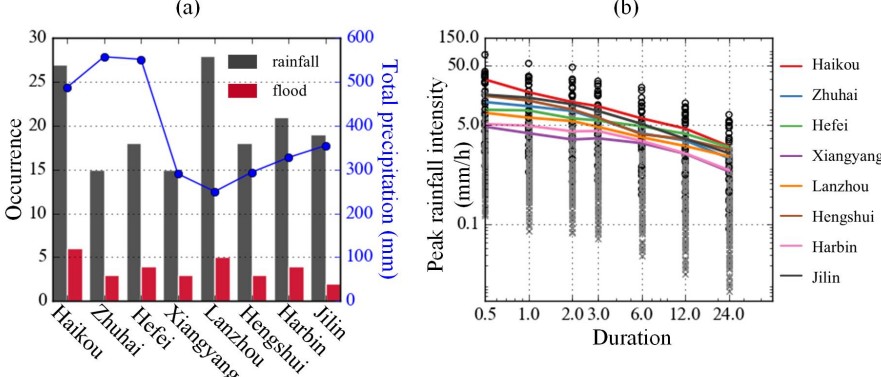

Figure 2. Total August precipitation and frequency of rainfall and city flooding events
(a). Variations in peak rainfall intensity (circles) and the flooding-triggering
precipitation threshold values (lines) that are derived from time windows ranging
from 0.5 to 24 hours (b).
**4.2 Normal rhythm of city**
The NLR records can serve as a proxy of the city residents' normal daily routine.
The normalized NLR show the eight cities have a similar diurnal rhythm (Fig. 3a). The
normalized NLR median climbs from a minimum at around 4:00 and to a peak right at
12:00. It starts to drop slightly and then peaks again at around 20:00. This general
pattern reflects the smartphone usage patterns of the city residents. Phone usage
starts to drop after the midnight when most residents start to rest. It reaches its first


peak during the lunch time as residents may request more LBS to find a place to eat.
After lunch time, phone usage remains at a high plateau, probably due to more LBS
requests for business purposes. Phone usage reaches the highest peak of the whole
day right after the normal work hours, indicating a significant increased need for the
LBS such as hailing nearby taxis to socialize with friends, go back home, or sending
geo-tagged posts for socializing.
The general diurnal pattern was superposed with subtle short-term NLR
variations. The NLR in the southern cities peaks and hits the bottom later at night
and before dawn, respectively, than that of the northern cities. This is very likely due
to the different lifestyles between the northern and southern residents in response
to the economic activities and day length. It is well-known that the southern China is
more active in economic and social activities and the southerners enjoy the night
activities more (Ma et al. 2019). By contrast, the northerners tend to end their
nightlife earlier and also become active earlier as the day breaks earlier in the north.
The total NLR is moderately correlated with the population of these cities (Fig.
3b). The 0.63 Pearson correlation coefficient (with a $p$ value of 0.046) indicates a
statistically significant positive relationship between the normalized NLR and the
population. As a result, we believe the NLR data could reflect the collective
geo-tagged behaviors of the city residents as a whole and consequently it could serve
as a proxy of the human responses to different environmental and social events.

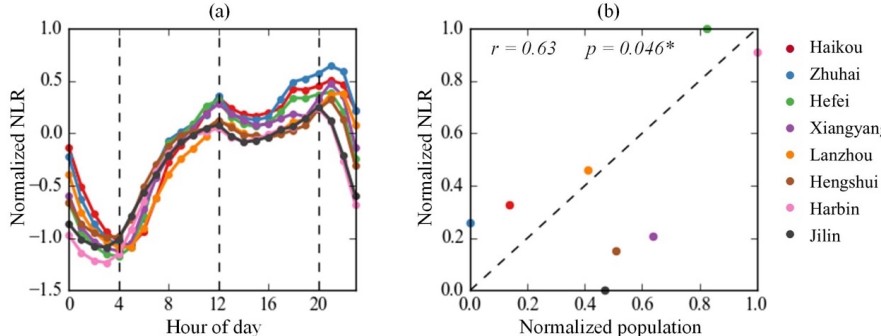

Figure 3. The diurnal variation patterns of the NLR in the eight cities (a) and a positive
correlation between the NLR and the total number of residents (b).

**4.3 Urban anomalies during rainstorms**
**4.3.1 City-scale analysis**
There are more positive than negative anomalies in the August time series
hourly NLR and most positive anomalies were found in pair with precipitation spikes
(Fig. 4). For example, two significant precipitation spikes in Harbin in the afternoon of




August 2nd and 3rd were closely associated with positive NLR anomalies. Few NLR
negative anomalies were identified in the eight cities except Zhuhai. This city was
significantly affected by Typhoon Hato, which brings huge amount of precipitation
and leads to a negative anomaly since the Afternoon of August 23rd in Zhuhai. Such a
significant negative anomaly could be attributed to serious communication
interruption or damages caused by the typhoon.
It is worthy to note that both positive and negative anomalies were also
identified when there is no rain in the cities. For example, two positive anomalies
were identified around August 28th in Harbin when there is no rain at all. The no-rain
anomalies must be triggered by other major events in the cities. However, at this
moment it is not easy to trace what local events may trigger such anomalies.
It is very interesting to notice that a couple of no-rain positive anomalies were
identified in the last week of August for almost all eight selected cities except Zhuhai.
These positive anomalies were obviously not associated with any special rainstorm
events. Instead, they are more likely to be associated with sort of national-wide
events, such as the college students' back to school and move-in events, which are
mainly scheduled in the last week of August every year in China. Such positive
anomalies were not found in Zhuhai, of which the 2017 back to school and move-in
events was postponed to the first week of October due to the significant damages
caused by Typhoon Hato. However, further studies, such as of the NLR of other cities
in China, are needed to consolidate this argument.

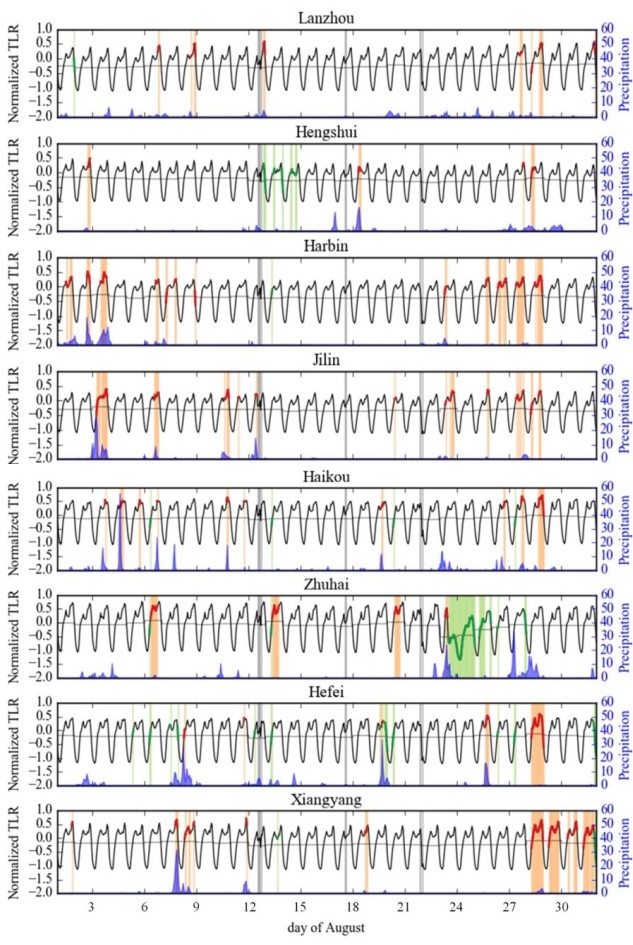

Figure 4. The time series NLR and rain events during August 2017. Positive and
negative anomalies were shown in orange and green colors, respectively. The light
gray columns show the periods when NLR data is missing.

6       We further quantitatively examined the association between rainfall events and

the NLR anomalies. Table 2 lists the $R_{pos}$ and $R_{neg}$, which are the ratios of the positive

and negative anomalies corresponding to the four scenarios (no rains, moderate,

heavy and violent rainstorm events) to the total number of anomalies identified over

the whole time series, respectively. As shown in Table 2, in total we identified 27, 19,

78, and 166 violent, heavy, moderate, and no rainstorm events in the eight cities,

respectively. Under different scenarios, the $R_{pos}$ is always higher than $R_{neg}$ except the

no rain scenario, in which there is no significant difference between these two ratios.

The rainstorm-related $R_{pos}$ increases from 0.22 to 0.70 as the rainstorms level up

from moderate to violent as compared to a no-rain $R_{pos}$ of 0.12. The rain-related or



no-rain $R_{neg}$ is no more than 0.22. The $R_{pos}$ is much higher than $R_{neg}$ when the cities
are affected by stronger rainfall events. For example, when the cities are affected by
violent storms, the $R_{pos}$ and $R_{neg}$ are 0.70 and 0.22 respectively. By contrast, the $R_{pos}$
and $R_{neg}$ are 0.22 and 0.06, respectively when the cities are affected by moderate
rainstorms. It is very likely that, when there are severe rainstorms, people may send
out more LBS requests in order to, for instance, search a route free of inundation
spots and less congested roads, order delivery food, or post geo-tagged photos of the
terrible moments.
A lower $R_{pos}$ of the heavy and moderate rainstorms may also be partly attributed
to the effect of data aggregation at the city scale. It is very common that a rainstorm
may influence only a part of the city and only lead to certain local positive anomalies.
In such a case, increase of the NLR in a small number of grids may not result in
significant changes of the NLR of the entire city and consequently no anomalies at
the city level. Analysis at the grid level, as reported in the next section, would show
how residents respond to the local rainstorm events.
The difference between the $R_{pos}$ and $R_{neg}$ also varies for different cities. For
example, the two violent rainstorms both trigger a positive anomaly in Xiangyang
and Harbin. By contrast, the five violent rainstorms in Zhuhai lead to the same
percent positive and negative anomalies. City Hefei is special. The same percent of
positive and negative anomalies are triggered by the five violent storms. However,
when Hefei is affected by the moderate and heavy rainstorms or even no rainfalls,
there are slightly more negative than positive anomalies.
Table 2. Numbers of different categories of rainstorms and the corresponding $R_{pos}$
and $R_{neg.}$

| Cities | No rainfall | | | Rainstorms | | | | | | | | |
|---|---|---|---|---|---|---|---|---|---|---|---|---|
| | | | | Moderate | | | Heavy | | | Violent | | |
| | N | $R_{pos}$ | $R_{neg}$ | N | $R_{pos}$ | $R_{neg}$ | N | $R_{pos}$ | $R_{neg}$ | N | $R_{pos}$ | $R_{neg}$ |
| Haikou | 27 | 0.04 | 0.22 | 14 | 0.21 | 0.00 | 3 | 0.33 | 0.00 | 8 | 0.75 | 0.00 |
| Zhuhai | 16 | 0.19 | 0.25 | 5 | 0.20 | 0.20 | 3 | 0.00 | 0.00 | 5 | 0.40 | 0.40 |
| Hefei | 19 | 0.05 | 0.32 | 7 | 0.00 | 0.14 | 2 | 0.50 | 1.00 | 5 | 0.60 | 0.60 |
| Xiangyang | 15 | 0.33 | 0.33 | 7 | 0.29 | 0.00 | 0 | - | - | 2 | 1.00 | 0.00 |
| Lanzhou | 29 | 0.07 | 0.10 | 17 | 0.24 | 0.06 | 5 | 0.20 | 0.20 | 0 | - | - |
| Hengshui | 19 | 0.00 | 0.21 | 11 | 0.18 | 0.09 | 2 | 0.00 | 0.00 | 2 | 0.50 | 0.00 |
| Harbin | 21 | 0.24 | 0.10 | 7 | 0.14 | 0.14 | 3 | 1.00 | 0.00 | 2 | 1.00 | 0.00 |
| Jilin | 20 | 0.15 | 0.15 | 10 | 0.40 | 0.00 | 1 | 1.00 | 0.00 | 3 | 1.00 | 0.33 |
| Overall | 166 | 0.12 | 0.20 | 78 | 0.22 | 0.06 | 19 | 0.37 | 0.11 | 27 | 0.70 | 0.22 |


**4.3.2 Grid-scale analysis: anomaly indices**
The S-H-ESD method was also used to detect the NLR anomalies at the grid level.
There are always more grids showing anomaly when the city was affected by a





rainstorm. Figure 5 provides an example to illustrate the grids with anomaly detected
during a rainstorm and the same time period in another day without rainfall in Jilin
and Haikou, respectively. Anomalies were identified in 56 grids in Jilin when it was hit
by a rainstorm at 7am on August 3, 2017. By contrast, anomalies are observed in only
10 grids during the same time period on August 6, 2017 when there is no rain at all.
In Haikou, anomalies are found in 52 grids during a rainstorm and only 19 grids when
there is no rain.

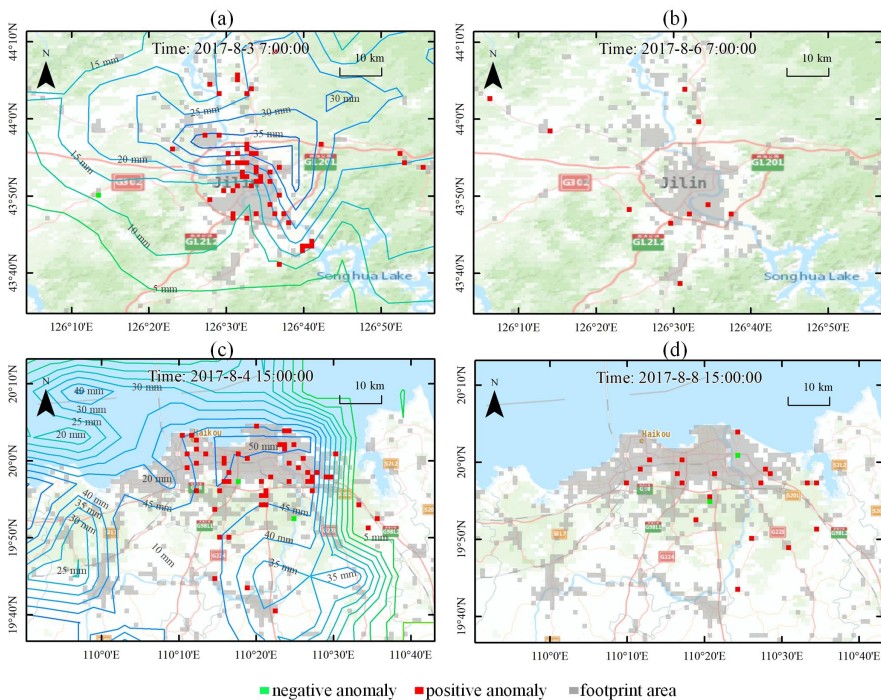

9        Figure 5. Grid with negative and positive anomalies within the footprint areas of

Haikou and Jilin. The contour lines show the precipitation.

12       The total number, total residual, and mean density of these anomalies were

then calculated (Fig. 6) for the cities when they were affected by flooding caused by a
typical rainstorm event (Table 3). The three anomaly indices show similar diurnal
variations as of the NLR diurnal rhythm and they all spiked to the level of an outlier
or even to an extreme value when the city was significantly affected by flooding




issues.
After the spikes, the anomaly indices usually bounce back to the same level
before for almost all the cities except Zhuhai, indicating most cities return to their
normal rhythms after the rainstorm interruption. However, Zhuhai was hit by the
category-3 Typhoon Hato at around 12:50 on August 23. The typhoon brought
intense rain, strong winds, and caused significant flooding issues and damages to the
city infrastructures, causing a sharp decline and persistent negative anomalies after
the landfall of Hato. It took more than 72 hours for the anomaly indices to bounce
back to the same level before Hato (not shown in Fig. 6).
Table 3. An exemplary flooding event in each of the cities.

| City | Urban flood event | Rainfall duration (h) | Accumulated precipitation (mm) | Half-hour peak rainfall intensity (mm/h) |
|---|---|---|---|---|
| Haikou | 8-4 15:00 | 10 | 117 | 77 |
| Zhuhai | 8-23 12:50 | 23 | 108 | 32 |
| Hefei | 8-25 17:00 | 13 | 72 | 25 |
| Xiangyang | 8-7 18:00 | 30.5 | 140 | 34 |
| Lanzhou | 8-12 21:00 | 9.5 | 14 | 5 |
| Hengshui | 8-18 08:00 | 15 | 67 | 18 |
| Harbin | 8-2 17:00 | 12.5 | 61 | 26 |
| Jilin | 8-3 07:00 | 38.5 | 185 | 31 |

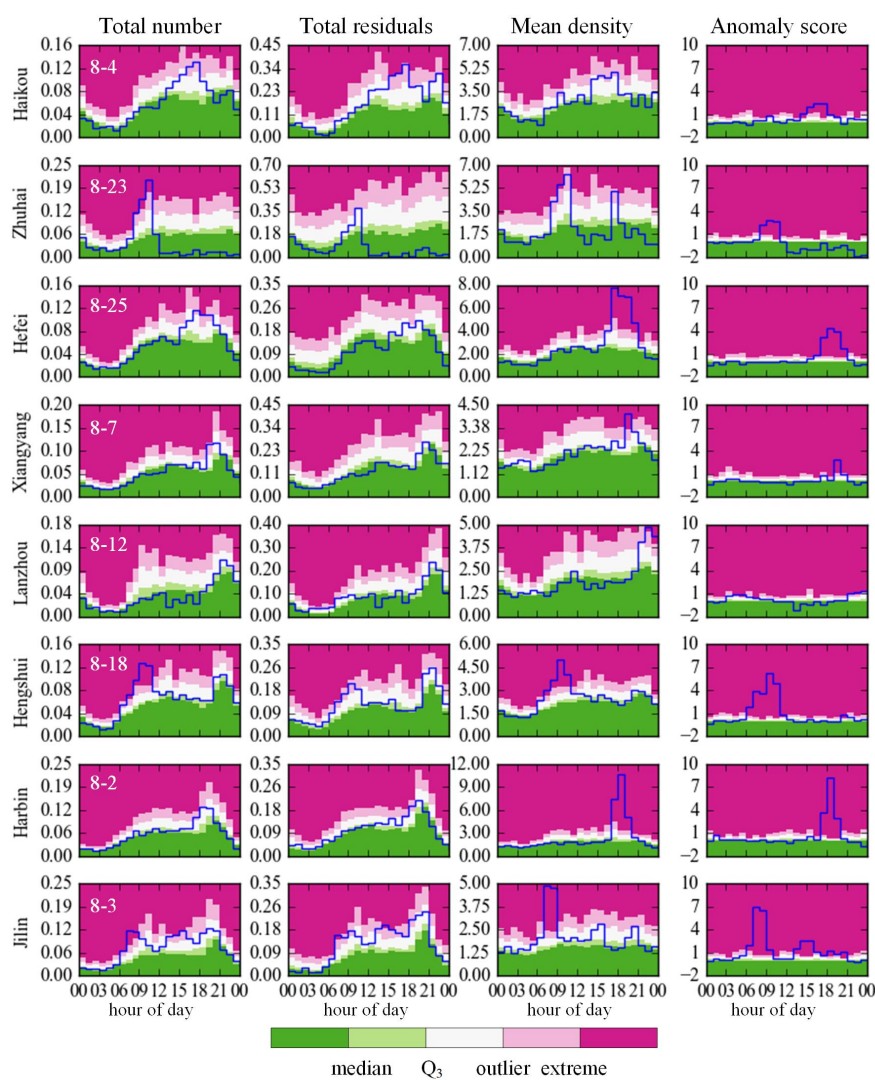

Figure 6. Intra-day variations in NLR, total residuals, mean density, and anomaly score

within 24 hours of a typical flooding event in each of the cities.

**4.3.3 Grid-scale analysis: anomaly score and rainfall intensity**

Given the anomaly score is indicative of the unusual responses of residents to

rainstorms, we further examined the relation between the anomaly score and the

rainfalls in these cities during the August 2017.


The grid-level $R_{pos}$ is much higher than the city-level counterpart with respect to
all types of events (Fig. 7a). Such a difference is mainly due to the different analysis
levels. We can easily identify the local anomalies per grid, which are more likely to be
obliterated at the city level due to the data aggregation. At the grid level, the $R_{pos}$ and
$R_{neg}$ also vary in response to the different levels of rainstorm events. All cities show a
higher $R_{pos}$ when they are affected by violent rainstorms (85%) than heavy rainstorms
(68%), in comparison with the $R_{pos}$ (56%) when the cities are not affected by any
rainfall events. However, the $R_{pos}$ of moderate rainstorms (45%) is less than the
no-rain $R_{pos}$, likely suggesting that low-intensity rainfall events may not necessarily
trigger NLR anomalies and other factors may contribute to the NLR anomalies at the
grid level.
How easily the rhythm of a city would be disrupted by a rainstorm is strongly
related to the anomaly-triggering peak rainfall intensity threshold (Fig. 7b), which
was calculated using the same the ideas in the methods developed by Cannon et al.
(2008) and Diakakis (2012). We plotted the peak rainfall intensity with respect to
whether there are anomalies or not for each city. The anomaly-triggering peak
rainfall intensity is defined as the upper limit of the rainfall intensity that tends to
lead to an NLR anomaly but actually not.
Every rainstorm with its peak intensity higher than the threshold would
definitely trigger an NLR anomaly. As a result, the cities with a lower threshold tend
to be more easily disrupted by a moderate or heavy rainstorm. For example,
Xiangyang has a very low threshold value of 1.4 mm/h. In August 2017, there are six
rainstorm events with peak rainfall intensity exceeding this threshold and they all
caused anomalies in this city.
However, even a rainstorm with its peak rainfall intensity below the threshold
may also trigger an NLR anomaly. For example, quite a few NLR anomalies were
found in Lanzhou, of which most rainstorms have its peak rainfall intensity below the
threshold (6.6 mm/h). This is because a heavy rainstorm at around 24:00 failed to
trigger an NLR anomaly as most people were sheltered at home and hence were not
affected. However, this rainstorm is included in the process to calculate the peak




rainfall intensity and increase the threshold. As a result, rainstorms with their peak
rainfall intensity below the threshold may also trigger anomalies, particularly in the
cities with more heavy and violent rainstorms after late night and before dawn.
The anomaly score is weakly correlated with rainfall intensity for each city (Fig.
7c). Three out of the eight cities (Harbin, Jilin, and Haikou) show a positive linear
relationship between the anomaly score and rainfall intensity. As the rainfall intensity
increases, the anomaly scores of the three cities increase linearly. Furthermore, the
slope coefficients of the correlations indicate how sensitive the rainfall intensity may
trigger anomalies. The city Harbin has the steepest slope thus slightly increase in
rainfall intensity would trigger anomalies more easily. By contrast, the gentlest slope
indicates Haikou is a city where the residents, in terms of their LBS request, are not
very sensitive to the increase of the rainfall intensity, probably because the residents
there are used to heave rains.
Around 31%, 23%, and 46% of the maximum anomaly scores were detected
before, at the same time, and after the rainfall intensity reaches its peak (Fig. 7d).
Specifically, 23%, 24%, and 20% of the anomaly score peaks simultaneously, within 1
hour, and within 2 hours of the rainfall intensity peaks, respectively. About 46% of
the anomaly score peaks after the rainfall intensity peaks, which is 50% more than
the number of the cases that anomaly score peaks ahead of the rainfall intensity
peak. As a result, we usually see the maximum positive anomalies (i.e. significant
disturbance in city rhythm) after the rainfall intensity reached a maximum value. It is
also possible for the anomaly to reach its peak before the peak of the rainfall
intensity if, for example, the cumulative rainfall is high enough to significantly impact
the city.


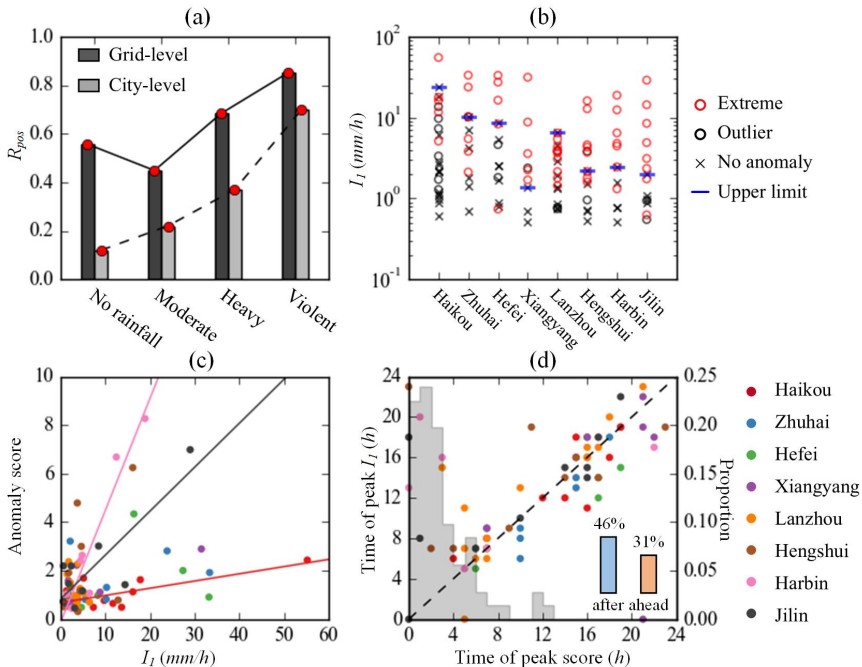

Figure 7. Correlation between peak rainfall intensity and anomaly score.

## 5. Conclusions

This study shows the potentials of the NLR data in reflecting city residents' collective geo-tagged behaviors. First of all, the NLR was moderately correlated with the population of the cities. Secondly, the time series NLR data well corresponds to the regular diurnal rhythm in all eight cities, which is characterized by limited activities from the midnight to early morning and very active LBS requests from noon to the evening. Thirdly, the time series NLR also reflects the different lifestyles in the northern and southern China, showing southerners enjoy late night life more whereas the northerners start their days earlier in the morning.

The anomalies of the NLR data are well with that the rainstorms, especially the violent ones, were very likely to trigger positive NLR anomalies at city level. At the grid level, the anomalies in response to rainstorms show a significant increase in the anomaly indices in terms of the total number, total residual, and mean density. The time series composite score derived from these three anomaly indices clearly shows




how city residents respond to rainstorms in terms of their LBS requests.

A same category rainstorm may not trigger NLR anomalies in the same way in every city. Essentially, the peak rainfall intensity of the rainstorms seems to be the key and such a threshold is significantly different among different cities. As a result, high peak rainfall intensity tends to trigger flooding and subsequently anomalies in the NLR data. Furthermore, the peak rainfall intensity is well associated with the peak anomaly score, further indicating it is the key factor that can trigger rainstorm-induced NLR anomalies.

It is worthy to note that other events may also contribute to NLR anomalies. There are a couple of positive anomalies in the last week of August for almost all cities except Zhuhai. The last week of August is the school registration time for college students in China. It is reasonable to expect such a nation-wide event may trigger NLR anomalies as shown in this study. However, some college cities may postpone the registration time and Zhuhai is one of them due to the significant damages caused by Typhoon Hato right before the registration week.

**Acknowledgements**

This research was funded by the National Key Research and Development Program of China (Grant Nos. 2017YFB0503605 and 2017YFC1503003), and the Strategic Priority Research Program of the Chinese Academy of Sciences (Grant No. XDA19040501). The IMERG data were provided by the NASA/Goddard Space Flight Center's PMM and PPS through http://pmm.nasa.gov/data-access/downloads/gpm (accessed 14 April 2019), and archived at the NASA GES DISC.

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
