# Peer review of "Urban anomalies in response to rainstorms based on smartphone location data: a"

_Natural Hazards and Earth System Sciences, 2019_

## Referee Comment (RC1) · Anonymous Referee #1 · 17 Jun 2019

general comments

The paper proposed a anomalies detection method based on smartphone location data. This method was tested in 8 cities in China and the authors claimed that strong association were found between NLR anomalies and rainstorms at city scale but in different cities, the association are different with their own characteristics. This paper is well organized and well written and I believe the geo-big data research community can benefit from this research, especially the rapid disaster management responding to real-time pattern from UGC. I advocate for the publication of this paper, however, some minor suggestions are followed.

[Figure]

specific comments

1. The smartphone location data needs to be explained further. As the most important data indicator in this study, readers need to know what exact service(s) provided by Tencent may generate location requests. In other words, a table including all Tencent's LBS helps readers infer the "underground" relationship between the anomaly scores and the storm events. 2. Is the correlation between peak rainfall intensity and anomaly score statistically significant in Figure 7(c)? This should be addressed. 3. The different association between rainfall events and the NLR anomalies should be explained. The impact by the government spending on urban infrastructure, such as drainage systems, as well as the climate zone at different cities can be mentioned in the discussion section.
* * *

---

## Referee Comment (RC2) · Anonymous Referee #2 · 23 Jul 2019

Comments

This paper used the S-H-ESD method to identify the anomalies of NLR at both the city and grid scale, which explored city residents' collective geo-tagged behaviors in response to rainstorm with the help of the NLR data generated by smartphone users. This paper is well organized and well written. Therefore, the research has obtained reliable and meaningful conclusions. I recommend for the publication of the paper. At the same time, some minor suggestions are followed:

1, Why did the author choose Tencent Big Data portal? The author need to briefly introduce the difference between Tencent big data platform and other platforms.

2, There are many advantages to using the NLR data. Meanwhile, is s there a disadvantage to using the NLR data? The author need to briefly introduce the disadvantage, and to trigger readers' thinking.

---

## Author Comment (AC1) · 1 Aug 2019

Revised manuscript is attached in supplement file.

Specific responses are as follows:

1. The smartphone location data needs to be explained further. As the most important data indicator in this study, readers need to know what exact service(s) provided by Tencent may generate location requests. In other words, a table including all Tencent's LBS helps readers infer the "underground" relationship between the anomaly scores and the storm events.

[Figure]

Response: Thanks for the comment. We included a table on page 5 to describe the common, though not all, applications that generate Tencent location requests.

2. Is the correlation between peak rainfall intensity and anomaly score statistically significant in Figure 7(c)? This should be addressed.

Response: Indeed. As shown by Fig. 7c, there are only three cities, Haikou, Harbin, and Jilin, showing linear relationship between peak rainfall intensity and anomaly score. Because for the three cities, the linear regression is statistically significant at the level of 0.05. We actually preformed the linear regression analysis for every city, however, the p-values for the other five cities are more than 0.05. So, there are no linear fit lines for them in Fig. 7c. Corresponding revision can be found on page 19 line 5-8

3. The different association between rainfall events and the NLR anomalies should be explained. The impact by the government spending on urban infrastructure, such as drainage systems, as well as the climate zone at different cities can be mentioned in the discussion section.

Response: Thanks for the comment. We added a short discussion on the possible connection with the urban infrastructure level and climatic condition on page 19 line 14-22.

Please also note the supplement to this comment:
https://www.nat-hazards-earth-syst-sci-discuss.net/nhess-2019-136/nhess-2019-136-AC1-supplement.pdf

**Supplement:**

[revised manuscript text omitted]

---

## Author Response (AR1)

Dear Professor Gregor,

Thanks for your comments. We carefully revised the manuscript according to the comments from you and the reviewers. The changes to the manuscript were marked and annotated in the uploaded pdf file. We also include our specific responses to the two reviewers as follows.

Reviewer 1:

*1. The smartphone location data needs to be explained further. As the most important data indicator in this study, readers need to know what exact service(s) provided by Tencent may generate location requests. In other words, a table including all Tencent' s LBS helps readers infer the "underground" relationship between the anomaly scores and the storm events.*

**Response:** We included a table on page 5 to describe the common, though not all, applications that generate Tencent location requests.

*2. Is the correlation between peak rainfall intensity and anomaly score statistically significant in Figure 7(c)? This should be addressed.*

**Response**: Indeed. As shown by Fig. 7c, there are only three cities, Haikou, Harbin, and Jilin, showing linear relationship between peak rainfall intensity and anomaly score. Because for the three cities, the linear regression is statistically significant at the level of 0.05. We actually preformed the linear regression analysis for every city, however, the p-values for the other five cities are more than 0.05. So, there are no linear fit lines for them in Fig. 7c. Corresponding revision can be found on page 19 line 5-8

*3. The different association between rainfall events and the NLR anomalies should be explained. The impact by the government spending on urban infrastructure, such as drainage systems, as well as the climate zone at different cities can be mentioned in the discussion section.*

**Response**: Thanks for the comment. We added a short discussion on the possible connection with the urban infrastructure level and climatic condition on page 19 line 14-22.

Reviewer 2:

1, *Why did the author choose Tencent Big Data portal? The author need to briefly introduce the difference between Tencent big data platform and other platforms.*

**Response**: Tencent has the largest social community in China. The 2018 annual report of Tencent wrote they have more than one billion monthly active users, which we believe the location request data generated by such a large group of users can provide good proxy for understanding human responses to rainstorms. Another reason for which we used this data source is that, the location request data are generated by users from multiple mobile apps (e.g. WeChat, QQ, DiDi, Meituan-Dianping, etc.). Such a large app ecosystem can capture more comprehensive user activity dataset than any single social platform. Please find our specific revision on page 5 line 7-12, and the Table 2 that listed the common apps.

2, *There are many advantages to using the NLR data. Meanwhile, is s there a disadvantage to using the NLR data? The author need to briefly introduce the disadvantage, and to trigger readers' thinking.*

**Response**: Yes, indeed. We added a short discussion on the limitation of the data on page 21 line 19-25.

Best regards,

Yunyan Du and all co-authors

---

## Author Response (AR2)

Dear Prof. Leckebusch,

Thank you for the revision comments. We carefully revised the manuscript point by point. Followings are our specific revisions and responses.

*1) Please give a full overview of applications using location-based services in Table 2 and which one are represented in which percentage in the LBS data?*
**Response**: We list and briefly explain the most widely used applications that generate location request data on Page 5 Lines 8 – 16. Tencent is a commercial company and doesn't release a complete list of the applications and their percentage in the location request data. Instead, Tencent only releases the total number of location request per square kilometer. We listed the number of monthly unique devices (MUD) for each app in Table 2. The MUD shows the number of active users of these apps.

*2) The title is not ideal: What is the variable that varies? Urban Anomalies is unclear. Anomalies of what?*
**Response**: We changed the title to "Anomalies of dwellers' collective geo-tagged behaviors in response to rainstorms: a case study of eight cities in China using smartphone location data", please see Page 1 Line 1 – 2.

*3) Abstract: Please explain more clearly the aim of the study at the beginning and give basic information which location-based services (e.g. map 20 navigation, car hailing, food delivery, etc.) are included in the NLRs.*
**Response**: We added three sentences to explain the aim of this study. Please see Page 1 Line 11 – 17.

Best regards
Yunyan Du and all co-authors